# Managed Surge Controller: A Docking Algorithm for a Non-Holonomic AUV (Sparus II) in the Presence of Ocean Currents for a Funnel-Shaped Docking Station

**DOI:** 10.3390/s23010241

**Published:** 2022-12-26

**Authors:** Joan Esteba, Patryk Cieślak, Narcís Palomeras, Pere Ridao

**Affiliations:** Computer Vision and Robotics Research Institute (VICOROB), University of Girona, 17003 Girona, Spain

**Keywords:** docking, AUV, ocean currents, non-holonomic

## Abstract

This paper presents a novel algorithm to dock a non-holonomic Autonomous Underwater Vehicle (AUV) into a funnel-shaped Docking Station (DS), in the presence of ocean currents. In a previous work, the authors have compared several docking algorithms through Monte Carlo simulations. In this paper, a new control algorithm is presented with a goal to improve over the previous ones to fulfil the specific needs of the ATLANTIS project. Performance of the new proposed algorithm has been compared with the results of the previous study, using the same environemnt on the Stonefish hardware-in-the-loop simulator.

## 1. Introduction

The AUV technologies presented a significant improvement during the last years. Several autonomous missions were developed in the field [1,2]. Nowadays, the common operating procedure of the AUVs is to deploy them (usually with a ship), develop the mission, and, finally, recover; in most of the cases during the same operational day. The natural next development for the consolidation of this technology is the creation of DSs, which can allow the AUVs to extend the operational time in the field. The DS have to offer protection, high-bandwidth communication channels, and the capacity to recharge the vehicle’s batteries. In the literature, several examples of DS systems can be found [3,4,5,6,7,8]. Each DS concept was tailored to a specific AUV and used its own perception and docking strategies. Several descriptive surveys about docking can already be found in the literature: [9,10,11].

Previously, the team from the University of Girona developed a prototype of a funnel-shaped DS that relied on an acoustic transponder and light beacons, in order to localize it from the vehicle [12]. The conclusions of those experiments have shown the need of developing a controller that considers the ocean currents. Moreover, in turbid water scenarios, the vision system was not appropriate. In the ATLANTIS project [13], one of the technologies that must be demonstrated is the semi-permanent deployment of the Sparus II AUV, using a docking station on the seabed. With this motivation, the authors have developed the following study.

In a previous publication [14], several algorithms were studied to cope with the autonomous docking, using a non-holonomic AUV, in the presence of ocean currents: the Pursuit Guidance with current compensation controller presented by [4,15], the Cross-track controller used in [3,16], the Fuzzy controller used in [17,18], the Touchdown alignment controller described in [19], the Sideslip controller based on [20], and the Sliding path controller described in [21]. None of the studied solutions offers satisfactory behavior when dealing with ocean current velocities higher than the docking velocity. Thus, a proposal of a novel control algorithm is presented in this work.

The paper is organized as follows: Section 2 presents the novel algorithm. Section 3 presents the experimental setup used in order to test the algorithm. Section 4 analyses the performance of the algorithm in different scenarios. Obtained results and the comparison with the other methods are presented in Section 5, discussed in Section 6, and finally, the conclusions are drawn in Section 7.

## 2. Proposal: Managed Surge Controller

This section presents the novel method to deal with the ocean currents controlling a non-holonomic AUV. The proposed method is called Managed Surge Controller (MSC).

### 2.1. Assumptions

#### 2.1.1. Assumption 1

We assume that the AUV motion is described using two degrees of freedom: surge and yaw. This is a consequence of the commonly adopted strategy of neglecting sway for under-actuated vehicles, maneuvering at low speed.

#### 2.1.2. Assumption 2

The inner loop controllers can track the desired surge velocity ($u_{d}$) and the desired yaw ($\psi_{d}$) with good accuracy.

#### 2.1.3. Assumption 3

The velocity of the ocean currents is constant, irrotational, and bounded.

#### 2.1.4. Assumption 4

The vehicle can measure its surge velocity, yaw angle, and the ocean current velocity.

#### 2.1.5. Assumption 5

The vehicle can measure its relative position with respect to the DS.

### 2.2. Concepts

Inspired by the analysis done in [14], the present method improves the results obtained with the previously studied methods because it can deal with ocean currents, the velocity of which are larger than the velocity of the docking of the AUV.

Figure 1 presents the basic variables involved in the process. Two different reference frames are presented, the {D} frame located at the position of the DS, and the {B} frame attached to the AUV body. The velocity in the {D} frame are symbolised by $\dot{x}$ and $\dot{y}$ with respect to the ground, and the velocities in the {B} frame are represented by *u* and *v* with respect to the water.

Figure 2 shows the velocities involved in the controller: (1) the ground speed (${}^{D}{\dot{\eta }}_{1B}={\left[\dot{x}\;\dot{y}\right]}^{T}$), and (2) the through water velocity (${}^{B}\nu_{1B}={\left[u\;v\right]}^{T}$). Finally, the ocean current vector (${}^{D}{\dot{\eta }}_{c}={\left[{\dot{x}}_{c}\;{\dot{y}}_{c}\right]}^{T}$) as well as the desired docking velocity (${}^{D}{\dot{\eta }}_{1D}={\left[{\dot{x}}_{D}\;0\right]}^{T}$) are represented in the inertial {D} frame.

#### 2.2.1. Model of the System

The kinematic system is represented by the following equations, recall Figure 1 and Figure 2:(1)$$\begin{array}{cc} {}^{D}{\dot{\eta }}_{1B} & ={}^{D}R_{B}\cdot {}^{D}\nu_{1B}+{}^{D}{\dot{\eta }}_{c} \end{array}$$
(2)$$\left[\begin{array}{c} \dot{x}\\ \dot{y} \end{array}\right]=\left[\begin{array}{cc} \cos(\psi ) & -\sin(\psi )\\ \sin(\psi ) & \cos(\psi ) \end{array}\right]\cdot \left[\begin{array}{c} u\\ v \end{array}\right]+\left[\begin{array}{c} {\dot{x}}_{c}\\ {\dot{y}}_{c} \end{array}\right]$$

This equation, assuming a negligible sway velocity (*Assumption 1* ), simplifies to:(3)$$\dot{x}=u\;\cos\left(\psi \right)+{\dot{x}}_{c},$$
(4)$$\dot{y}=u\;\sin\left(\psi \right)+{\dot{y}}_{c}.$$

#### 2.2.2. Docking Scenarios

According to Figure 3, three docking scenarios can be defined, depending on the robot (${}^{D}\nu_{1}$), current (${}^{D}\nu_{c}$) and docking (${}^{D}\nu_{1_{D}}$) velocities in the $x_{D}$ axis:Scenario A (${\dot{x}}_{c}\leq 0$): The current opposes to the robot speed. Therefore, a higher through water robot speed is required to achieve the desired inertial docking velocity.Scenario B (${\dot{x}}_{c}>0$ and ${\dot{x}}_{c}<{\dot{x}}_{D}$): The current speed, being smaller than the desired docking velocity, adds to the through water robot velocity to achieve the inertial docking speed.Scenario C (${\dot{x}}_{c}\geq {\dot{x}}_{D}$): A current speed higher than the docking velocity, requires a backward through-water robot velocity to achieve the desired inertial docking velocity.

In scenarios A and B the surge velocity will normally be positive and the AUV heading will be opposite to ${\dot{y}}_{c}$. In contrast, in scenario C, the surge velocity will normally be negative and the heading will be in the direction of ${\dot{y}}_{c}$.

#### 2.2.3. Crab Angle

To be able to compensate the lateral ocean current (${\dot{y}}_{c}$) with a non-holonomic AUV it is necessary to use a crab angle ($\psi_{c}$). This crab angle has the goal of aligning the robot to the axis of the DS ($\dot{y}=0$), while keeping the desired docking velocity (${}^{D}{\dot{\eta }}_{1D}$). Therefore (Equation 2) can be rewritten as:(5)$$\left[\begin{array}{c} {\dot{x}}_{D}\\ 0 \end{array}\right]=\left[\begin{array}{cc} \cos(\psi_{c}) & -\sin(\psi_{c})\\ \sin(\psi_{c}) & \cos(\psi_{c}) \end{array}\right]\left[\begin{array}{c} u\\ 0 \end{array}\right]+\left[\begin{array}{c} {\dot{x}}_{c}\\ {\dot{y}}_{c} \end{array}\right].$$

Solving the system, the crab angle can be expressed as:(6)$$\tan\left(\psi_{c}\right)=\frac{-{\dot{y}}_{c}}{{\dot{x}}_{D}-{\dot{x}}_{c}},$$
assuming ${\dot{x}}_{D}-{\dot{x}}_{c}\neq 0$.

#### 2.2.4. Entrance Problem

The problem of making a torpedo-shaped AUV enter a funnel-shaped DS can be modeled with the simplification that the DS is represented by an isosceles triangle and the AUV by a straight line directed along its main axis (see Figure 4).

If the symmetry of the system is taken into account, there are only three successful docking scenarios (Figure 5):Scenario I: It represents the ideal entrance, where the AUV enters in a straight line with the same heading as the DS and aligned with its origin DS.Scenario II: The robot heading is not aligned to the $x_{D}$ axis, but misaligned to the right.Scenario III: The robot heading is not aligned to the $x_{D}$ axis, but misaligned to the left.

In Scenario II and III, the AUV completes the docking thanks to the geometrical properties of the system. Both scenarios differ in the energy lost during the collision, Scenario III being the one with the highest losses. Following the analysis of energy lost during the collision (reported in [14]), the AUV should try to perform Scenario I if possible, targeting Scenario II alternatively, and using Scenario III as the last resort. It is worth noting that Scenario I is not easy to perform with Sparus II AUV, in presence of ocean currents, due to its non-holonomic nature.

#### 2.2.5. Path

A crab angle is needed to compensate for the ocean currents, therefore the method creates a path parallel to the DS axis at an appropriate distance from it, to take advantage of the torpedo-like shape of the AUV and the funnel-shaped DS, see Figure 6.

The path is created calculating a gap ($y_{g}$), with respect to the axis of the DS using the crab angle, the DS and the AUV geometry. In order to calculate this gap, let us consider the geometry of the problem when the AUV reaches the DS in scenarios II and III (Figure 7). For the entrance in Scenario II, the gap can be calculated as:(7)$$y_{g,II}=\left(F+l/2\right)\sin\left(-\psi_{c}\right).$$

The maximum crab angle admissible to enter to the DS in Scenario II is computed using the simplified funnel shape:(8)$$\psi_{c_{II}}=\mathrm{atan}\left(F_{y}/F_{x}\right),$$

This allows to calculate the maximum gap as:(9)$$y_{g}=\left(F+l/2\right)\sin\left(\mathrm{atan}\left(F_{y}/F_{x}\right)\right).$$

For crab angles larger than $\psi_{c_{II}}$, the system needs to perform the entrance Scenario III. The maximum crab angle for the entrance Scenario III, in case of $2\;F_{y}<l/2$ (that fulfils the mechanic characteristics of the system presented), can be calculated as (see Figure 8, and recall Figure 5):(10)$$\psi_{c_{III}}=\frac{\pi }{2}-\mathrm{atan}\left(F_{y}/F_{x}\right),$$

in order to guarantee that the AUV does not hit a disfavoured part of the DS. In this case the $y_{g_{III}}$ is fixed to be equal to $F_{y}/2$ (see Figure 7). In summary, (Equation 11) sets the value of $y_{g}$ for all the cases as:(11)$$y_{g}=\left\{\begin{array}{cc} -\left(F+l/2\right)\sin\left(\psi_{c}\right), & -\psi_{c_{II}}\leq \psi_{c}\leq \psi_{c_{II}}\\ -\mathrm{sign}\left(\psi_{c}\right)\;F_{y}/2, & |\psi_{c}|>\psi_{c_{II}}. \end{array}\right.$$

#### 2.2.6. Path Following

The Sparus II AUV has direct control over the surge velocity (i.e., *u*) as well as over the heading (i.e., ψ). Let the cross-track path error be defined as:(12)$$e=y-y_{g}$$

With this notation, the objectives of the path following the controller are to achieve:(13)$$\lim_{t\to \infty }e\left(t\right)=0$$
(14)$$\lim_{t\to \infty }\psi_{d}\left(t\right)=\psi_{c}$$
(15)$$\lim_{t\to \infty }u_{d}\left(t\right)=\frac{{\dot{x}}_{D}-{\dot{x}}_{c}}{\cos(\psi_{c})},$$

### 2.3. Control Law

The control law regulates the desired heading ($\psi_{d}$) and the desired surge ($u_{d}$). The desired heading is defined as:(16)$$\psi_{d}=\psi_{c}+\beta ,$$
where β is a correction term depending on the look-ahead distance (Δ) and the cross-track error (*e*), see Figure 9:(17)$$\tan\left(\beta \right)=-\;\frac{e}{\Delta },$$
where Δ is defined by a constant value ($k_{\Delta }>0$) and a sign criteria (Equation 21):(18)$$\Delta =\frac{k_{\Delta }}{\mathrm{sign}({\dot{x}}_{ss})},$$
being ${\dot{x}}_{ss}$ the through water velocities of the AUV in the steady state in the {D} frame (i.e., when *e* is zero).

The desired surge velocity is given by:(19)$$u_{d}=\frac{{\dot{x}}_{ss}}{\cos(\psi )}-c,$$
where *c* is a correction term, introduced to adjust the response of the system:(20)$$c=k_{1}\;\mathrm{atan}\left(k_{2}\;e\right)\;\mathrm{sign}\left(\psi \right);k_{1},k_{2}>0$$

The correction modifies the basic velocity representation from Figure 2 into the one shown in Figure 10. The steady state inertial velocities can be calculated as:(21)$${\dot{x}}_{ss}=\dot{x}-{\dot{x}}_{c},$$
(22)$${\dot{y}}_{ss}=-{\dot{y}}_{c},$$
where $\dot{x}$ is set as ${\dot{x}}_{D}$.

#### 2.3.1. State Space Formulation

The system evolution is represented by the cross-track error (Equation 12), which, according to (Equation 4), has the following time derivative:(23)$$\dot{e}=u\;\sin\left(\psi \right)+\;{\dot{y}}_{c}.$$

Now, assuming $u=u_{d}$ and $\psi =\psi_{d}$, the error dynamics are given by:(24)$$\dot{e}=\left[\frac{{\dot{x}}_{ss}}{\cos(\psi_{d})}-k_{1}\;\mathrm{atan}\left(k_{2}\;e\right)\;\mathrm{sign}\left(\psi_{d}\right)\right]\sin\left(\psi_{d}\right)+{\dot{y}}_{c}$$

#### 2.3.2. Equilibrium Points

An equilibrium point is reached when $\dot{e}=0$, i.e., when $\psi_{d}=\psi_{c}$. If this condition is applied to (Equation 24), it follows that $e_{eq}=0$ if ${\dot{x}}_{c}\neq {\dot{x}}_{D}$, Δ≠0, $k_{1}>0$, and $k_{2}>0$. In order to fulfill this condition, and because ${\dot{x}}_{D}$ is a value that we can set, in the case of having ${\dot{x}}_{D}={\dot{x}}_{c}$ the ${\dot{x}}_{D}$ will be increased according to:(25)$${\dot{x}}_{D}=\left\{\begin{array}{cc} {\dot{x}}_{D}, & {\dot{x}}_{ss}\in R\setminus \left(-0.2,0.2\right)\\ {\dot{x}}_{c}+0.2, & {\dot{x}}_{ss}\in \left[0.0,0.2\right)\\ {\dot{x}}_{c}-0.2, & {\dot{x}}_{ss}\in \left(-0.2,0.0\right) \end{array}\right.$$

#### 2.3.3. Setting the Gains

The definition of $k_{1}$ and $k_{2}$ corresponds to the maximum velocity and acceleration that the correction *c* (Equation 20) will impose on the system. Considering the characteristics of the Sparus II, a maximum correction velocity of 0.7m/s, and a maximum correction acceleration of $0.5\;{m/s}^{2}$ are desirable.

In order to set $k_{1}$, the function y=atan(x) is analysed. This function has a horizontal asymptote at y=π/2 and in y=−π/2. Taking into account the performance of Sparus II, the horizontal asymptote of (Equation 20) must be set to y=0.7m/s, consequently $k_{1}=0.7\cdot 2/\pi \approx 0.4456\;m/s$. To compute the maximum rate of change of the correction, both $k_{2}$ and $k_{\Delta }$ must be known. For this reason, in a first step $k_{2}$ is set as $1\;m^{-1}$.

The parameter $k_{\Delta }$ is computed in the Appendix A, assuming $k_{2}=1\;m^{-1}$, in order to fulfil the stability conditions formulated in Section 2.3.5.

#### 2.3.4. Domain of the Controller

In order to evaluate the controller presented in this paper, a certain domain is set:(26)$$\left\{\begin{array}{c} X=\left\{{\dot{x}}_{c}\;|\;-0.5\leq {\dot{x}}_{c}\leq 0.5\right\}\left[m/s\right]\\ Y=\left\{{\dot{y}}_{c}\;|\;-0.5\leq {\dot{y}}_{c}\leq 0.5\right\}\left[m/s\right]\\ E=\left\{e\;|\;-10\leq e\leq 10\right\}\left[m\right] \end{array}\right.$$

Ocean current velocities must be within the range of those that the Sparus II AUV can withstand. Since the docking maneuver begins after a homing process, we can ensure that the cross-track error of the AUV position belongs to the stability domain. If during the maneuver, the cross-track error falls outside of the domain, the docking maneuver is cancelled and the whole process is repeated.

#### 2.3.5. Stability

In order to demonstrate the stability of the system using the Lyapunov Direct Method, the following Lyapunov candidate is proposed:(27)$$V\left(e\right)=\frac{1}{2}\;e^{2},$$
that fulfills the first and second Lyapunov conditions: V(0)=0, and V(e)>0∀e≠0.

The first order time derivative of the Lyapunov candidate can be expressed as:(28)$$\dot{V}=e\;\dot{e}.$$

To demonstrate exponential stability, we show that the system fulfills the condition (following [22]):(29)$$\dot{V}\leq -\lambda \;V,$$
for some λ>0. The mathematical proof of stability is given in the Appendix A.

#### 2.3.6. Maximum Acceleration Verification

In Section 2.3.3, a $k_{2}=1\;m^{-1}$ was assumed to be able to calculate a $k_{\Delta }$ that fulfils the stability conditions. In order to verify that the obtained gains do not result in surpassing the assumed capabilities of the Sparus II, for the presented controller, an optimization problem was formulated. In the problem, a maximum rate of change of the correction (*c*) is searched for, to check that it is not exceeding the maximum assumed acceleration of the robot $\dot{c}=0.5\;{m/s}^{2}$. First, we simplify (Equation 20) as:(30)$$c^{\prime }=k_{1}\mathrm{atan}\left(k_{2}e\right),$$
to avoid deriving the signum function, and because the maximum value of $\dot{c}$ is not affected. The rate of change of $c^{\prime }$ is given by:(31)$${\dot{c}}^{\prime }=\frac{k_{1}k_{2}}{k_{2}^{2}e^{2}+1}\dot{e}.$$

The optimization problem is formulated as follows:(32)$$\begin{array}{cc} \max_{{\dot{x}}_{c},{\dot{y}}_{c},e}\; & {\dot{c}}^{\prime }\\ s.t.\; & {\dot{x}}_{c}\in X\\ \; & {\dot{y}}_{c}\in Y\\ \; & e\in E \end{array}$$

Considering the domain set for the controller (Section 2.3.4), the parameters:(33)$$\left\{\begin{array}{c} {\dot{x}}_{D}=0.3\;m/s\\ k_{1}=0.4456\;m/s\\ k_{2}=1\;m^{-1}, \end{array}\right.$$
and together with (Equation 25), it can be deduced that three optimization problems must be solved for three subsets of the ${\dot{x}}_{c}$ domain: $X_{1}=\left[-0.5,0.1\right]$, $X_{2}=\left[0.1,0.3\right]$ and $X_{3}=\left[0.3,0.5\right]$, and the maximum of these three problems is the solution in the whole domain (Equation 26). The problem (Equation 32) was solved using the IPOPT [23] solver and yielded a result of (0.44979 m/s^2^).

#### 2.3.7. Minimum Docking Distance

With the control laws defined, the minimum necessary distance to successfully perform docking, from a kinematic point of view, can be calculated, by solving (Equation 24). The domain of Section 2.3.4 is considered together with the follwoing parameters:(34)$$\left\{\begin{array}{c} {\dot{x}}_{D}=0.3\;m/s\\ k_{1}=0.4456\;m/s\\ k_{2}=1\;m^{-1}\\ k_{\Delta }=6\;m, \end{array}\right.$$
applying the correction presented in (Equation 25). The maximum time to reach the equilibrium (|e|≤0.05m) can be calculated. This time applied to the desired docking velocity gives an approximation of the minimum necessary docking distance:(35)$$D_{min}={\dot{x}}_{D}\;t.$$

The minimum distance necessary to dock, in the worst case scenario, is close to 25 m. Note that the maximum velocity and acceleration in yaw have not been taken into account, because (for the set parameters) it has a low influence when the working yaw angle is reached. Note also that this is a controller defined for the docking maneuver (starting at ±10m in $y_{D}$, with a heading favourable to the DS, i.e., −1.5>ψ<1.5), the homing maneuver will require its own additional distance.

## 3. Experimental Setup

In order to do a consistent comparison, the same experimental setup, as presented in [14], was used.

### 3.1. Hardware

The non-holonomic **auv!** used for this test was the torpedo-shaped Sparus II [24,25], see Figure 11. The AUV comes equipped with three thrusters, one vertical and a pair of horizontal, allowing for control in the surge, heave, and yaw. The control system supports inputs in force, velocity, and position. In this study we chose to control the vehicle in velocity, for the surge and heave, and in position, for the yaw.

The funnel-shaped DS developed by the Univeristy of Girona [12] (see Figure 12) was represented in simulation.

### 3.2. Simulation

In this research an advanced open-source marine robotics simulator, called Stonefish [26], was used. Full dynamics and hydrodynamics of Sparus II were simulated, including ocean current influence, together with a complete suite of its sensors, see Figure 13. Moreover, the docking station model was recreated in the simulation, with a high attention to detail, allowing for realistic assessment of docking performance. Specifically for this research, the simulator was extended to support acoustic communication and positioning devices. More details can be found in the previous work [14].

## 4. Performance

The objective of this section is to show the performance of the algorithm in the simulated scenario. The concept utilized to develop the high-level controller presented in this paper is to use the strong features of the Sparus II, to achieve maximum possible performance. The Sparus II AUV, as a non-holonomic robot without a rudder, requires a combined action of its two horizontal thrusters, in order to control the heading. This fact added to the non-symmetric behavior of the thrusters makes a notably lower response in the heading input than in the surge velocity.

The controller presented utilizes mainly the surge in order to correct the position of the AUV and reduce *e*, see Figure 14.

### 4.1. Docking Scenarios A and B

For the docking scenarios A and B, see Figure 3, the heading of the AUV is opposite to ${\dot{y}}_{c}$, it can be seen in the video [27]. It can be understood, analyzing the velocity vectors, i.e., ${\dot{x}}_{D}$ is bigger than ${\dot{x}}_{c}$, that the surge velocity has to be positive; this being the case (and taking into consideration that we have a non-holonomic AUV), the surge has to point in the opposite direction to the ${\dot{y}}_{c}$, in order to be able to compensate for it.

A set of initial conditions are simulated (Equation 36), for kinematics (solving (Equation 3) and (Equation 4)), and plotted in Figure 15 and Figure 16. They are also simulated in dynamics (using Stonefish) and plotted in Figure 17 in order to compare both results.
(36)$$\left\{\begin{array}{c} {\dot{x}}_{D}=0.3\;m/s\\ {\dot{x}}_{c}=0.2\;m/s\\ {\dot{y}}_{c}=0.2\;m/s\\ e\in [-10:2:10]\;m\\ k_{1}=0.4456\;m/s\\ k_{2}=1\;m^{-1}\\ k_{\Delta }=6\;m, \end{array}\right.$$

### 4.2. Docking Scenario C

For the docking scenario C, the heading of the AUV is the same as the direction of ${\dot{x}}_{c}$, see the video [28]. Again, analyzing the velocity vectors, i.e., ${\dot{x}}_{D}$ is smaller than ${\dot{x}}_{c}$, the surge velocity has to be negative, so the surge has to point in the same direction as the ${\dot{y}}_{c}$, in order to be able to compensate for it.

A set of initial conditions are simulated (Equation 37) for kinematics (solving (Equation 3) and (Equation 4)) and plotted in Figure 18 and Figure 19. They are also simulated in dynamics (using Stonefish) and plotted in Figure 20 in order to compare both results.
(37)$$\left\{\begin{array}{c} {\dot{x}}_{D}=0.3\;m/s\\ {\dot{x}}_{c}=0.5\;m/s\;\\ {\dot{y}}_{c}=0.5\;m/s\\ e\in [-10:2:10]\;m\\ k_{1}=0.4456\;m/s\\ k_{2}=1\;m^{-1}\\ k_{\Delta }=6\;m, \end{array}\right.$$

### 4.3. Entrance Scenario II

An example of the performance achieved in the entrance scenario II can be found in [29], see Figure 5.

### 4.4. Entrance Scenario III

An example of the performance achieved in the entrance scenario III can be found in [30], see Figure 5.

## 5. Results

In order to compare the new algorithm with the state-of-the-art, the data obtained in the article [14] are used. In this previous article, different algorithms, already published in the literature, were compared at different levels. Being that level 3 is the most representative for the authors’ needs, the data from this level are used to compare with the proposed algorithm, since they were tested at the same level.

As it was presented in [14], in order to estimate the quality of the docking process in a funnel-shaped DS, a novel technique based on the geometrical analysis of the entrance of the AUV was used. With this technique, we can evaluate the methods with a ’score’ that ranges from 0 to 1, with 1 being the perfect docking.

In order to represent the comparison, a Boxplot is presented in Figure 21.

The **3d!** plot of the results is presented in Figure 22, and the **2d!** plot in Figure 23.

A table with the numerical results compared can be seen in Table 1.

## 6. Discussion

The Managed Surge Controller achieved a mean score of 0.891, being the highest value for the proposed conditions. The controller is able to dock the robot in all the ocean current conditions tested in the previous study (e.g., from −0.4 m/s to 0.4 m/s on both axis). As it can be seen in Figure 21, it not only has the best mean scores, but also a low standard deviation, showing that the results are consistent.

As the score indicates and as can be further appreciated in Figure 22 and Figure 23, a good performance in all the ocean current conditions tested during the exercise was achieved. Ocean current velocity values larger than the ones studied in the previous paper were also studied in simulation, also presenting good results. However, it is not recommended to operate the Sparus II in more harsh conditions.

The Sparus II is a torpedo-shaped AUV equipped with three thrusters. It is designed to perform extended surveys, where the precision of the position is not crucial, implying that it is not capable of performing complicated maneuvers with high precision. One of the facts that is not reflected in the score is the simplicity of the maneuver, from a practical point of view. The simpler the maneuver, the easier it is to perform it, using the real vehicle. In the docking maneuver, the **msc!** (**msc!**) follows a straight-line, focusing on utilizing the horizontal thrusters optimally to work against the ocean current forces and correcting the cross-track error.

If the **msc!** is compared with the Sliding path controller (presented in [21] and implemented in [14]), that achieved a mean score of 0.790, one of the main differences is the developed maneuver. The Sliding path controller implies a notable precision in the maneuver to be able to enter exactly in the desired position, which is hard to achieve using the Sparus II. It is also not capable of minimizing the cross-track error when the ocean current velocity is favorable and close to or larger than the docking velocity, when it is in the approaching path. However, during the sliding path, it still can compensate for the error.

## 7. Conclusions

This paper has presented a novel controller to dock with a non-holonomic AUV, in a funnel-shaped DS, dealing with ocean currents. The paper proves the stability of the controller and the the expected behavior is further confirmed using a very realistic dynamic simulator. In a previous work [14], after exhaustive survey, several algorithms to face the same problem were implemented and tested in the context of the ATLANTIS project, using the Stonefish simulator. This previous work concluded with a problem to study: how to minimize the cross-track error when the ocean current velocity is favorable and close to or larger than the docking velocity. In this article, the authors have developed an algorithm capable of dealing with this problem while maintaining or improving the rest of the evaluated criteria.

The novel proposal was compared with the state-of-the-art algorithms, with the criteria developed in [14], presenting the best results. In future work, this controller will be implemented in Sparus II and tested in real scenarios. Also, a new funnel-shaped DS will be designed and built, in order to meet the requirements of the ATLANTIS project. The authors expect to be able to achieve docking without the use of vision systems; and, if it is the case, with the presence of ocean currents in the context of the project.

## Figures and Tables

**Figure 1 sensors-23-00241-f001:**
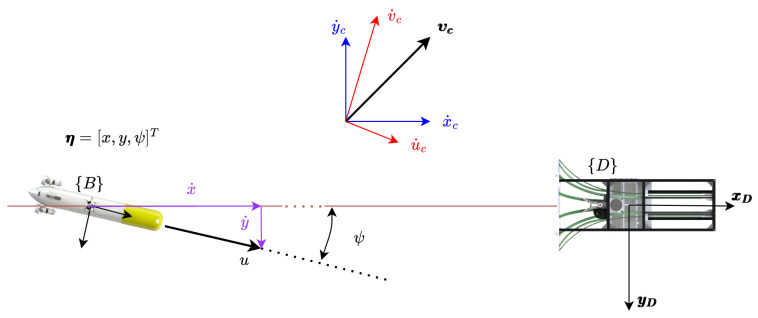
Geometrical representation of basic variables. Two different reference frames are presented, the {D} frame located at the position of the DS, and the {B} frame attached to the AUV body. The velocity in the {D} frame are symbolised by $\dot{x}$ and $\dot{y}$ with respect to the ground, and the velocities in the {B} frame are, with respect to the water, represented by *u* and *v*.

**Figure 2 sensors-23-00241-f002:**
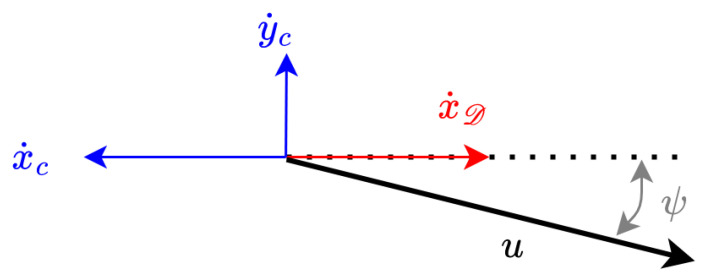
Geometrical representation of basic velocities.

**Figure 3 sensors-23-00241-f003:**
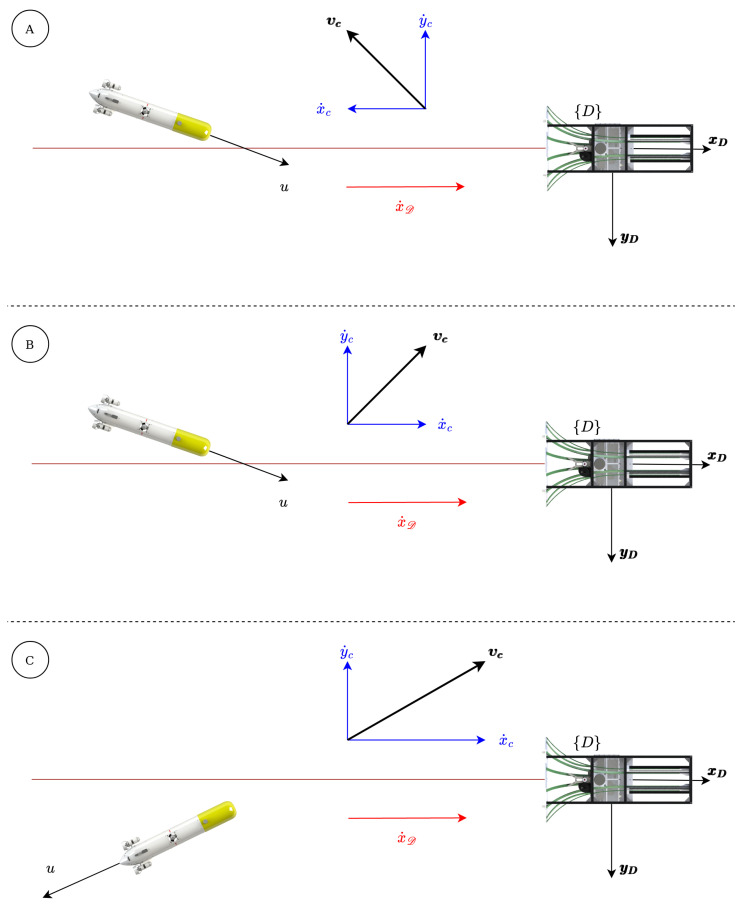
Possible docking scenarios.

**Figure 4 sensors-23-00241-f004:**
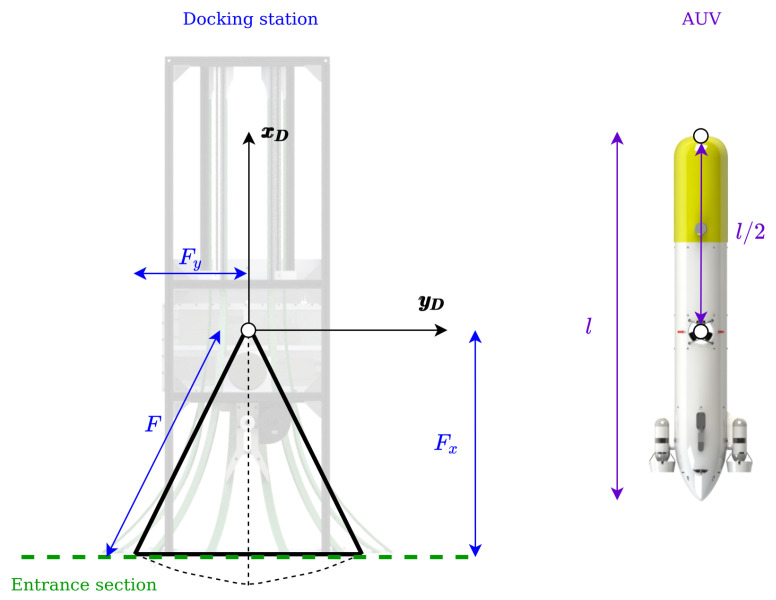
Geometrical problem simplification.

**Figure 5 sensors-23-00241-f005:**
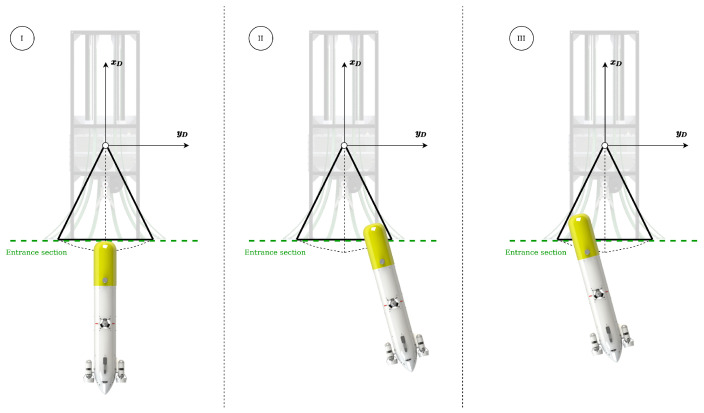
Successful entrance scenarios.

**Figure 6 sensors-23-00241-f006:**
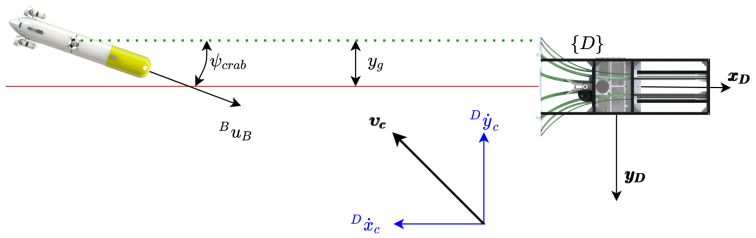
Path concept representation.

**Figure 7 sensors-23-00241-f007:**
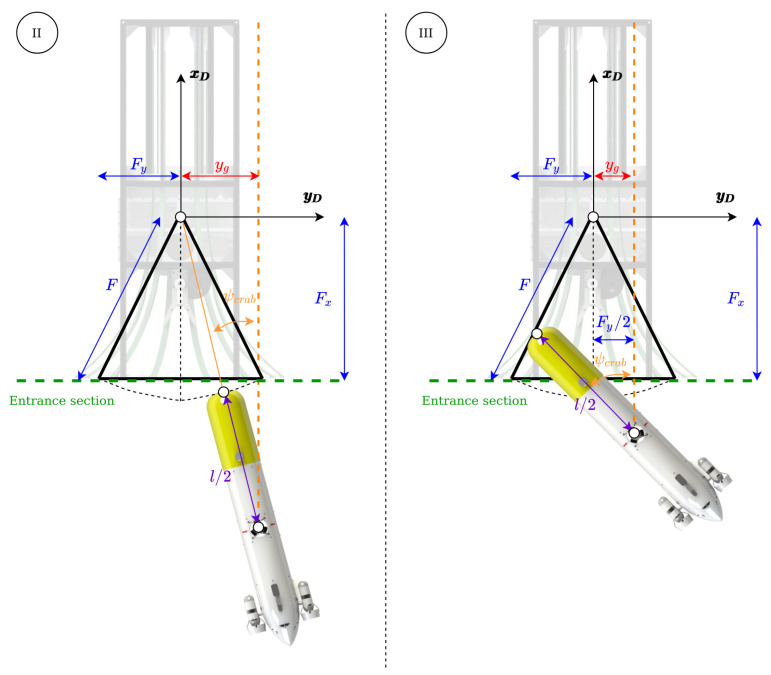
Gap calculus concept.

**Figure 8 sensors-23-00241-f008:**
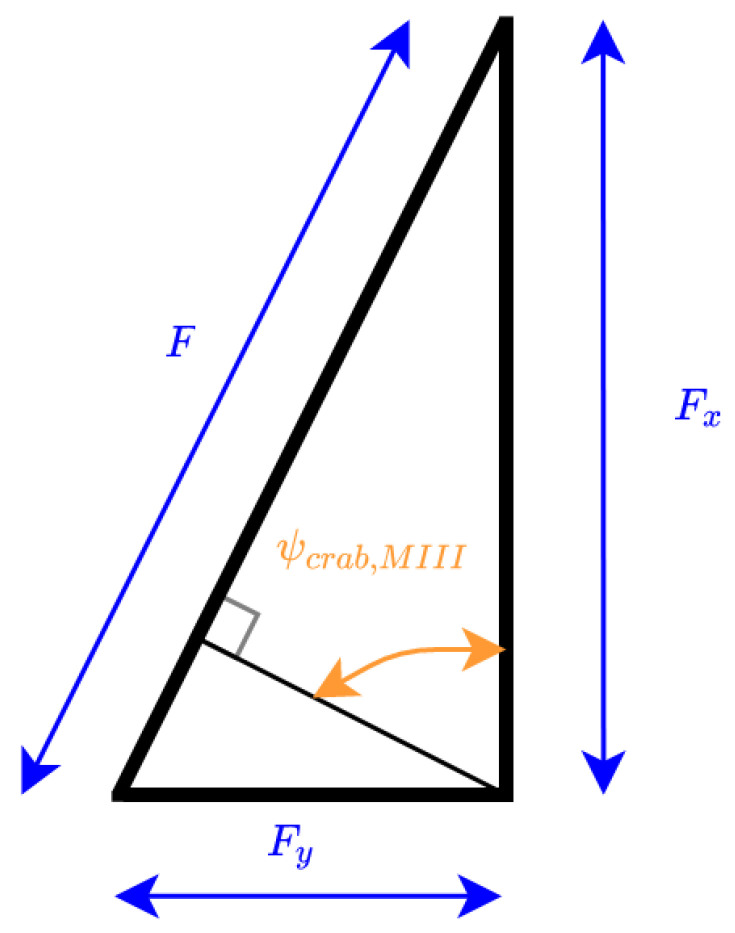
Maximum entrance Scenario III gap calculus concept.

**Figure 9 sensors-23-00241-f009:**
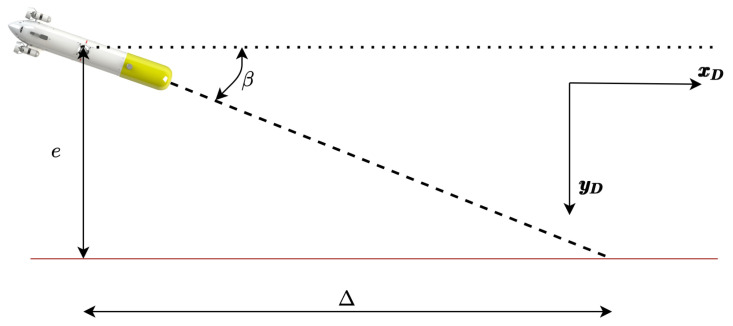
Beta concept representation.

**Figure 10 sensors-23-00241-f010:**
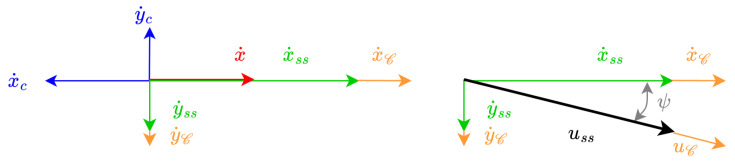
Geometrical representation of corrected velocities. Here, ${\dot{x}}_{ss}$, and ${\dot{y}}_{ss}$ are through water velocities of the AUV in the steady state in the {D} frame (i.e., when *e* is zero); $u_{ss}$ the through water velocity of the AUV in the steady state in the {D} frame; and ${\dot{x}}_{C}$, ${\dot{y}}_{C}$, and $u_{C}$ the velocities due to *c*.

**Figure 11 sensors-23-00241-f011:**
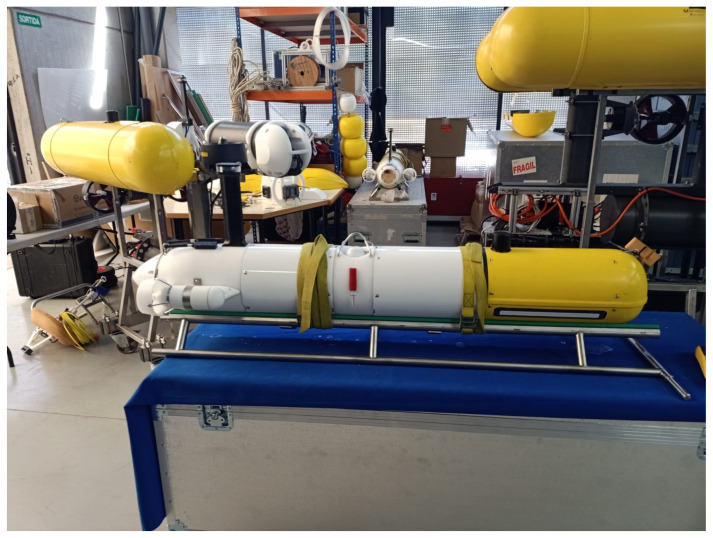
Photography of the Sparus II.

**Figure 12 sensors-23-00241-f012:**
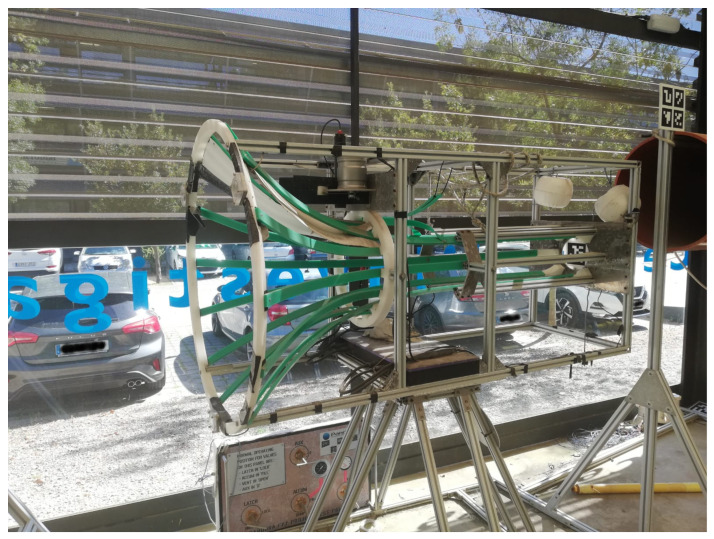
Photography of the docking station.

**Figure 13 sensors-23-00241-f013:**
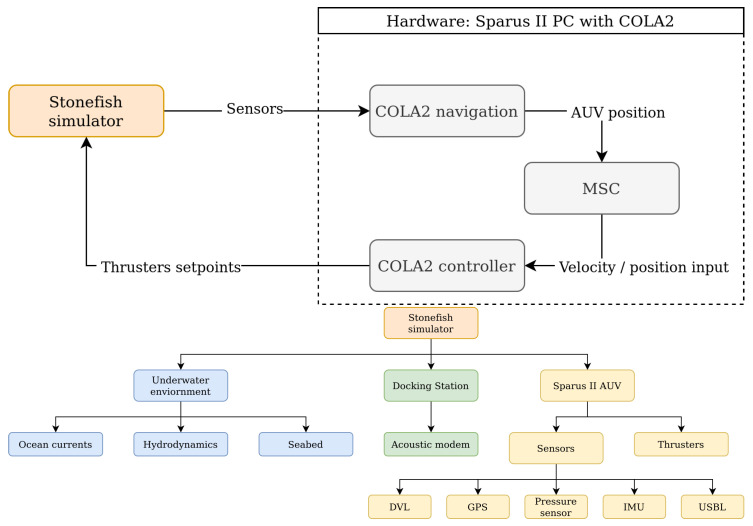
Conceptual representation of the hardware-in-the-loop simulation. The Sparus II architecture is in communication with the Stonefish simulator, which disposes of a model of the DS, the AUV with its sensors and thrusters [25], and the representation of the underwater environment.

**Figure 14 sensors-23-00241-f014:**
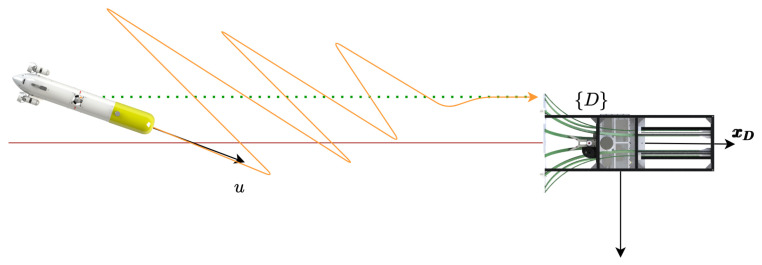
Conceptual performance representation.

**Figure 15 sensors-23-00241-f015:**
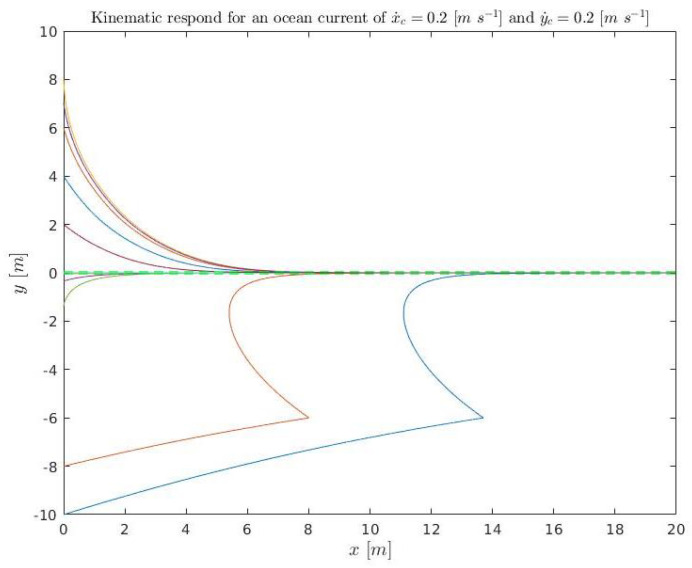
Kinematic simulation for the conditions (Equation 36) relative to position. The colored lines represent the different simulations and the green discontinuous line the acceptance tolerance.

**Figure 16 sensors-23-00241-f016:**
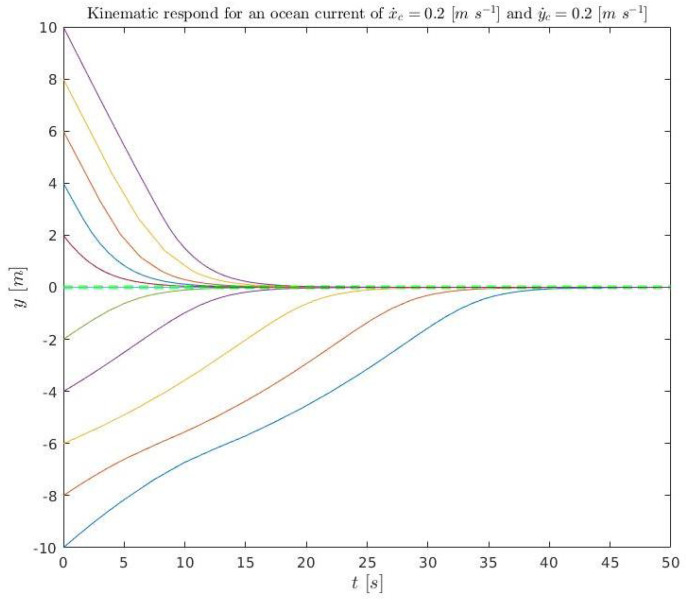
Kinematic simulation for the conditions (Equation 36) relative to time. The colored lines represent the different simulations and the green discontinuous line the acceptance tolerance.

**Figure 17 sensors-23-00241-f017:**
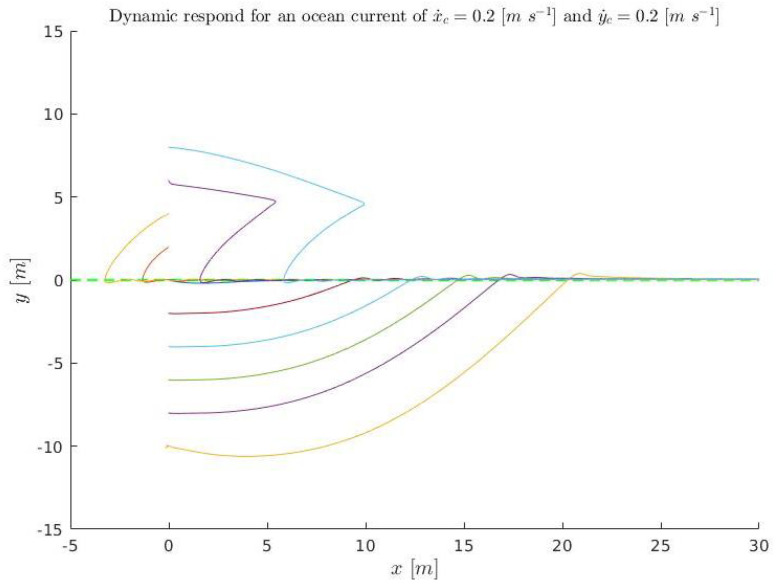
Dynamic simulation for the conditions (Equation 36). The colored lines represent the different simulations and the green discontinuous line the acceptance tolerance.

**Figure 18 sensors-23-00241-f018:**
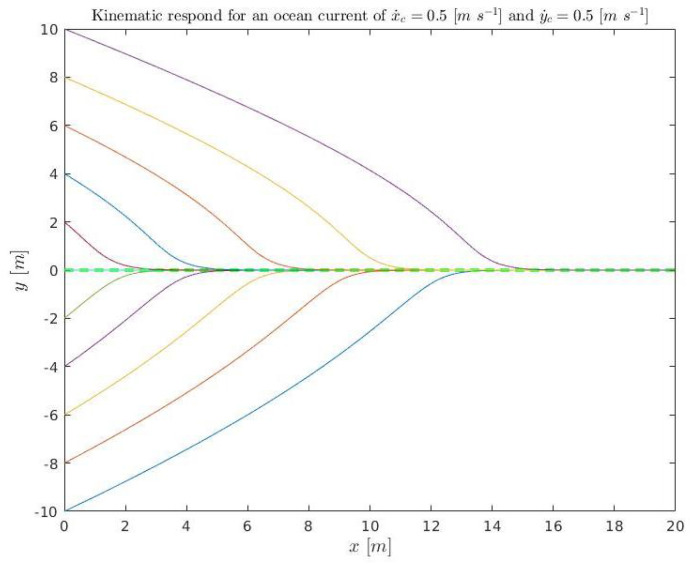
Kinematic simulation for the conditions (Equation 37) relative to position. The colored lines represent the different simulations and the green discontinuous line the acceptance tolerance.

**Figure 19 sensors-23-00241-f019:**
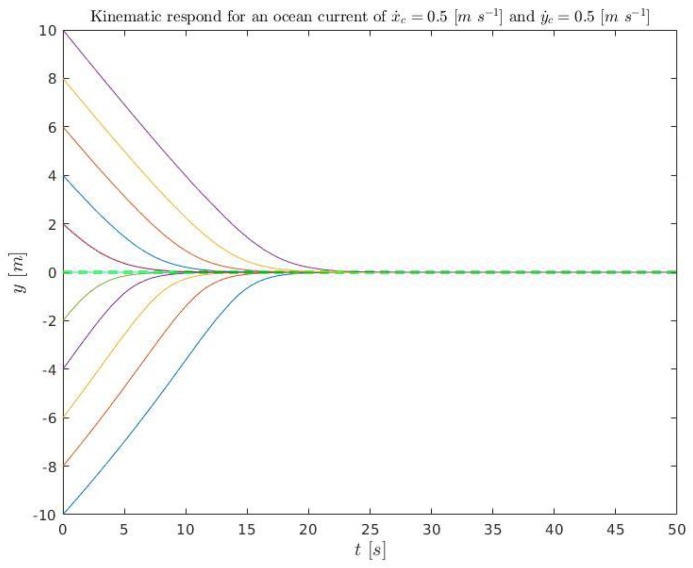
Kinematic simulation for the conditions (Equation 37) relative to time. The colored lines represent the different simulations and the green discontinuous line the acceptance tolerance.

**Figure 20 sensors-23-00241-f020:**
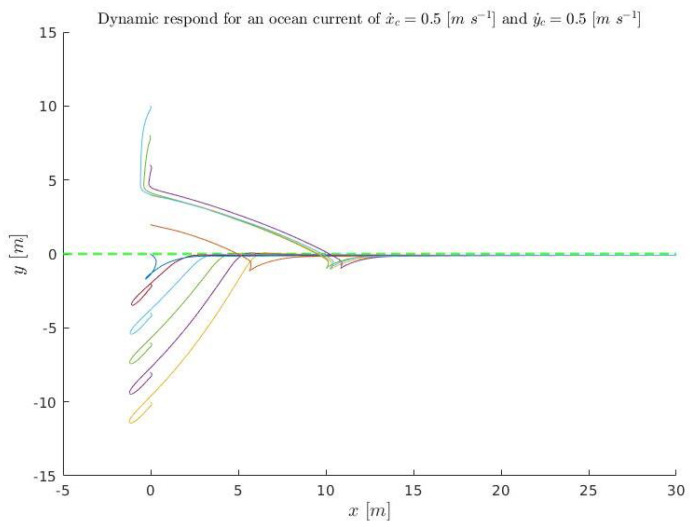
Dynamic simulation for the conditions (Equation 37). The color line represent the different simulations and the green discontinuous line the acceptance tolerance.

**Figure 21 sensors-23-00241-f021:**
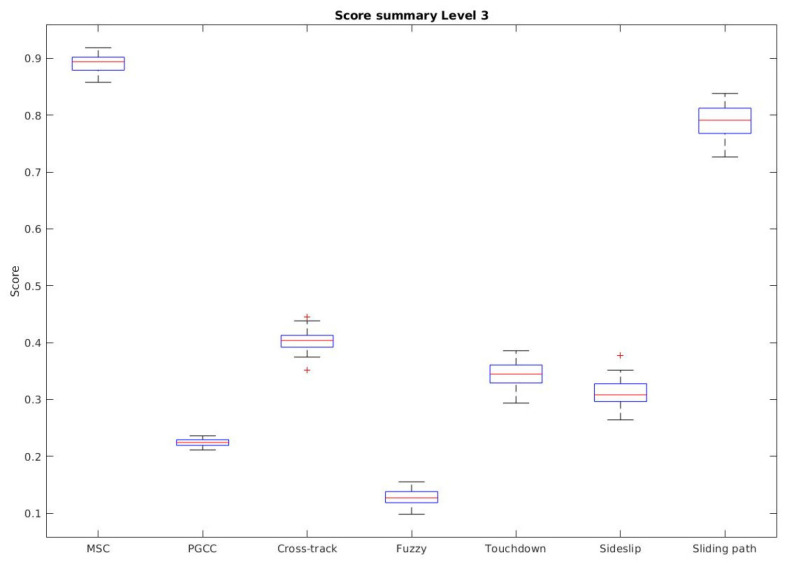
Score summary comparison between the different methods, taking the results of [14], using a box plot.

**Figure 22 sensors-23-00241-f022:**
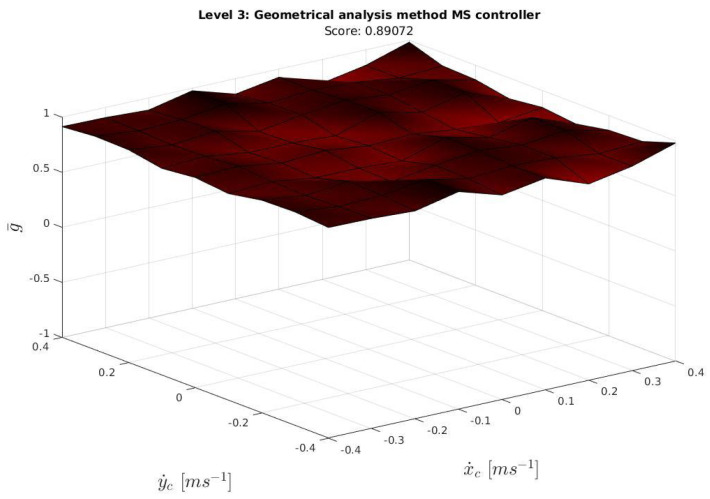
3D Geometrical analysis for the Managed surge controller.

**Figure 23 sensors-23-00241-f023:**
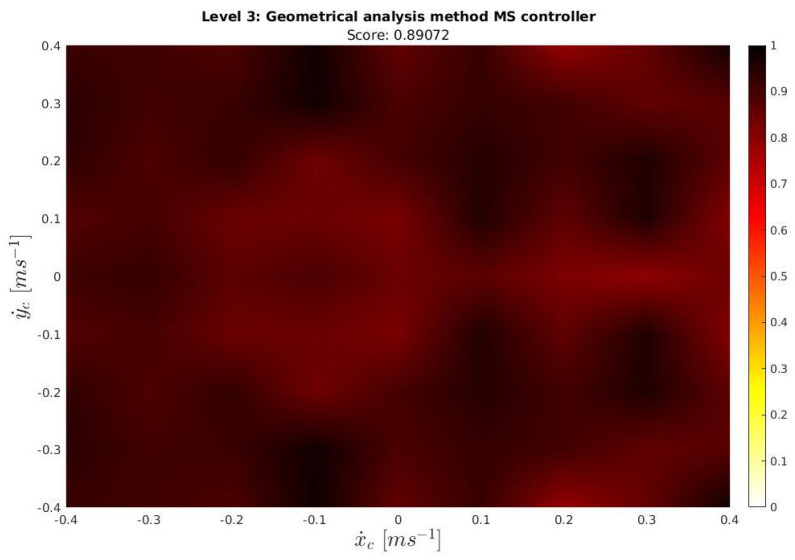
2D Geometrical analysis for the Managed surge controller.

**Table 1 sensors-23-00241-t001:** Score results comparison between the different methods, taking the results of the level 3 of [14].

Method	Score	
	Mean	Std
Managed surge controller	0.891	0.013
PGCC controller	0.224	0.008
Cross-track controller	0.403	0.019
Fuzzy controller	0.128	0.014
Touchdown alignment controller	0.345	0.022
Sideslip controller	0.311	0.021
Sliding path controller	0.790	0.025

## Data Availability

The data used is already published in this paper.

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
