# Peer review of "Managed Surge Controller: A Docking Algorithm for a Non-Holonomic AUV (Sparus II) in the Presence of Ocean Currents for a Funnel-Shaped Docking Station"

_sensors, 2022, doi:10.3390/s23010241_

Round 1

Reviewer 1 Report

The  comments and suggestions for authors are attached in the file

Author Response

1. Comment 1

Thank you for your appreciation, we applied your suggestion.

2. Comment 2

Added the explanation from the text to the description of the Figure:

Two different reference frames are presented, the \(\{D\}\) frame located at the position of the \ac{ds}, and the \(\{B\}\) frame attached to the \ac{auv} body. The velocity in the \(\{D\}\) frame are with respect to the ground symbolised by \(\dot{x}\) and \(\dot{y}\), and the velocities in the \(\{B\}\) frame are with respect to the water represented by \(u\) and \(v\).

We decided to not add a superindex in the variables with the frame to avoid too large expressions.

3. Comment 3

Added Figure 13, following your suggestion.

4. Comment 4

Modification applied, thank you.

5. Comment 5

Checked again, and removed two notations, thank you.

6. Comment 6

We discussed (and tried) if it is possible to modify the cited figure in order to make it significantly different from its ancestor, but we reached the conclusion that it will decrease the understanding of the rest of the paper. We really appreciate your suggestion, and we hope that you can understand it.

7. Comment 7

Modified the conclusions with the following text:
The novel proposal was compared with the state of the art algorithms, with the criteria developed in \cite{Esteba2021}, presenting the best results. In future work, this controller will be implemented in Sparus II and tested in real scenarios. Also, a new funnel-shaped \ac{ds} will be designed and built, in order to meet the requirements of the ATLANTIS project. The authors expect to be able to achieve docking without the use of vision systems, and if it is the case with the presence of ocean currents in the context of the project.

We upload the paper updated with your suggestions, thank you so much.

Reviewer 2 Report

The paper is very good overall but for one minor issue. There are a number of self citations for the authors, 12 I counted. There are two conditions that should be met for self citation, first is relevance... how does this work use prior work to build on or expand efforts. Second condition is the citations cannot loop, the no citation should cite another paper with the authors that has already been cited in the one citation. In other words, one citation and the others can cite like a chain but not multiple times in the various papers.

Here is an example of relevance: "Several autonomous missions were developed in the field [1 4]." This is grouping of 4 citations with no link to the statement stated. Also, in the field I think you mean autonomous vehicle research field. A sentence should contain a cogent idea that ties to one of the citations. 4 are not required and 2 of which are from the authors of this paper. This is an academic peer reviewed paper and we want to maintain the standards of academic integrity. You can use one self reference here and show how prior work leads to the effort described in this paper. There shouldn't be more than one self reference unless you can justify why another is required.

Author Response

Thank you for your appreciation. We removed the referred citations as you suggested. 

We would like to clarify that this work is a continuation of several works already developed in our laboratory (“standing on the shoulders of giants”). We are using the Stonefish simulator, the Cola2 architecture, the previous Docking Station, and the Sparus II AUV. We hope that this justifies the number of citations.  Thank you again for your comprehension.

We upload the paper updated with your suggestions, thank you so much.

Round 2

Reviewer 2 Report

Thank you for cleaning up the references. If this is built on prior work you can use one reference and explain how this is building on that effort. I believe I mentioned this... the explanation is important. Otherwise I recommending the paper be accepted.